# Rapid Eye Movement Sleep during Early Life: A Comprehensive Narrative Review

**DOI:** 10.3390/ijerph192013101

**Published:** 2022-10-12

**Authors:** Hai-Lin Chen, Jin-Xian Gao, Yu-Nong Chen, Jun-Fan Xie, Yu-Ping Xie, Karen Spruyt, Jian-Sheng Lin, Yu-Feng Shao, Yi-Ping Hou

**Affiliations:** 1Departments of Neuroscience, Anatomy, Histology, and Embryology, Key Laboratory of Preclinical Study for New Drugs of Gansu Province, School of Basic Medical Sciences, Lanzhou University, 199 Donggang Xi Road, Lanzhou 730000, China; 2Sleep Medicine Center of Gansu Provincial Hospital, Lanzhou 730000, China; 3Université de Paris, NeuroDiderot–INSERM, 75019 Paris, France; 4Integrative Physiology of the Brain Arousal Systems, CRNL, INSERM U1028-CNRS UMR 5292, University Claude Bernard Lyon 1, Centre Hospitalier Le Vinatier–Neurocampus Michel Jouvet, 95 Boulevard Pinel, CEDEX, 69675 Bron, France; 5Key Lab of Neurology of Gansu Province, Lanzhou University, Lanzhou 730000, China

**Keywords:** rapid eye movement (REM) sleep, sleep ontogeny, neurodevelopmental disorders, infant, childhood

## Abstract

The ontogenetic sleep hypothesis suggested that rapid eye movement (REM) sleep is ontogenetically primitive. Namely, REM sleep plays an imperative role in the maturation of the central nervous system. In coincidence with a rapidly developing brain during the early period of life, a remarkably large amount of REM sleep has been identified in numerous behavioral and polysomnographic studies across species. The abundant REM sleep appears to serve to optimize a cerebral state suitable for homeostasis and inherent neuronal activities favorable to brain maturation, ranging from neuronal differentiation, migration, and myelination to synaptic formation and elimination. Progressively more studies in Mammalia have provided the underlying mechanisms involved in some REM sleep-related disorders (e.g., narcolepsy, autism, attention deficit hyperactivity disorder (ADHD)). We summarize the remarkable alterations of polysomnographic, behavioral, and physiological characteristics in humans and Mammalia. Through a comprehensive review, we offer a hybrid of animal and human findings, demonstrating that early-life REM sleep disturbances constitute a common feature of many neurodevelopmental disorders. Our review may assist and promote investigations of the underlying mechanisms, functions, and neurodevelopmental diseases involved in REM sleep during early life.

## 1. Introduction

Rapid eye movements (REM) during sleep were first reported in adult humans in 1953 [1]. This sleep state associated with REM was then termed REM sleep by William Dement [2,3,4]. Human REM sleep was subsequently found to be associated with vivid dreaming [4,5], cortical EEG desynchronization, loss of muscle tone [4,6], penile erections, and fluctuation of autonomic systems [6,7,8].

Next, an assessment of the proportions of REM sleep in various age periods demonstrated remarkably similar findings in humans [9,10,11,12,13] and animals [14,15,16,17,18,19,20,21,22,23,24], i.e., a higher percentage of REM sleep in neonates than in adults. In other words, across species, REM sleep during the early development of life (in this paper, we defined early life in humans, rats, and cats as 38 gestational weeks (GW) to 2 years, postnatal day 0 (P0) to P30, and P0 to P45, respectively) is remarkably abundant. More recently, it is thought that REM sleep provides a frequently activated brain state during this critical maturational period. It allows adequate and inherent neuronal activities favorable to brain maturation, ranging from neuronal differentiation, migration, and myelination to synapse formation and elimination [25,26]. It equally plays a critical role in the plasticity of the developing brain [26,27,28]. Blumberg et al. concluded that more myoclonic twitches of skeletal muscles occurring during early life REM sleep trigger sensory feedback and therefore contribute to the establishment of the sensorimotor system [27,29,30]. The ontogenetic sleep hypothesis pointed out that the dramatic decline in REM sleep amounts across development manifests that REM sleep ontogenesis is a remarkably conserved feature of mammalian sleep [28], which suggested that REM sleep is ontogenetically primitive.

This review focuses on the characteristics of REM sleep in early life, some REM sleep-related disorders in humans, and their underlying mechanisms as examined in animals. We aim to bring REM sleep back into the spotlight, and particularly to foster the potential of translational research through our cross-species approach.

## 2. REM Sleep in Early Development of Humans

### 2.1. REM Sleep Amount across Early Development

Between 28 and 30 weeks of gestation, most a fetus’s time is spent in REM sleep, with little signs of an NREM sleep state [13,31,32]. Thus, along with the gestational age, REM sleep is progressively reduced from 80% at 30 weeks to 67% between 33 and 35 weeks, and further to 58% between 36 and 38 weeks.

REM sleep and NREM sleep in newborn infants are also known respectively as active sleep (AS) and quiet sleep (QS) [11,33,34,35,36,37,38]. Indeterminate sleep, namely when the exact sleep state cannot be distinguished, is also observed during the early phase of life [11,33,34].

Full-term newborns spend one-third of their day and one-half of their sleep in REM sleep [9,10,11,12,38]. Preterm infants have less sleep but more frequent REM episodes. Later, the percentage in REM sleep over total sleep time (TST) progressively declines with age and reaches a roughly stable proportion of 20% in REM sleep and 80% in NREM sleep at about the age of three years to remain approximately constant throughout childhood, adolescence, and early adulthood [9,11,12,39,40], whereas in late adulthood, it does decline slightly [41].

### 2.2. Polysomnographic, Behavioral and Physiological Characteristics of REM Sleep in Early Development

#### 2.2.1. Polysomnographic Features

In humans, sleep and electroencephalographic (EEG) patterns of REM and NREM sleep can be difficult to decipher before 30 weeks of gestation. They become constant by 36 to 38 weeks of gestation [13,31,34,37,42]. Yet their classification remains complex, and their EEG remains atypical during this period of two weeks given the precocious developmental stage [13,34,35].

The sleep states in full-term newborns are discerned by EEG, electro-oculograms (EOG), and electromyograms (EMG). EEG during periods of REM sleep is a low-voltage, relatively fast activity, sometimes appearing a little slower than the waking EEG. Meanwhile, EOG invariably appears as single or clustered high-amplitude bursting waves of rapid eye movements. On EMG, notably, phasic muscular contractions in the background of the absence of resting muscle activity during REM sleep are numerously traced. EEG during NREM sleep, in contrast, is characterized by high-voltage slow waves without eye movement tracings and phasic muscular contractions [12,35,43,44].

Another characteristic EEG pattern observed in newborn infants is that unlike the normal adult pattern in which NREM sleep precedes always REM sleep, REM sleep directly succeeds waking episodes at sleep onset [12,45,46], similar to the occurrence of sleep-onset REM sleep (SOREMS) in narcoleptic patients [47,48]. With increasing age, the EEG patterns during REM sleep show a progressive increase in frequency and amplitude. The average duration of these patterns decreases from around 25 and 30 min at 2 and 4 weeks of age to about 16 min at 16–24 weeks [49].

The mature stage 2 of NREM sleep with EEG spindles emerges between 6 and 9 weeks, and slow delta waves mixed with theta frequencies appear at approximately 12 weeks of age [11,50,51]. K-complex begins to have a drastic increase in the percentage of stage 2 of NREM sleep over TST by the end of 6 months.

Stage 3 of NREM sleep tends to occur during the nocturnal hours and peaks in the early period of the night at 4–6 months of age [50]. The percentage of sleep periods beginning with REM sleep declines with age, and REM/NREM cycles lengthen. Infants at 3 weeks of age have 60% REM sleep onset, however, those whose sleep at 6 months begins with REM sleep are reduced to 20% [46].

#### 2.2.2. Behavioral and Physiological Features

Other conspicuous behavioral features in early human life associated with the EEG pattern of REM sleep are eye movements, local muscle contractions, facial appearances, and body movements [12,13,34,35,38,52,53], which is why this state is also called active sleep [35,38,53].

Eye movements are very rare before the 28th gestational week, and their number remains lower in preterm than in full-term newborns [35,54]. However, rapid eye movements in the form of clusters start to increase after birth, reaching a plateau at about 4 months [55].

Fetuses between 38 and 40 weeks of gestational age show a large number of spontaneous body movements during REM sleep, though fewer than those among newborns [53]. REM sleep atonia becomes obvious after the 40th gestational week [56]. During REM sleep, term newborns frequently display, in addition to rapid eye movement bursts; grimaces; small weak cries; smiles; twitches of the face and extremities; and brief athetoid writhing of the torso, limbs, and digits. Some facial mimicries may resemble the appearance of sophisticated expressions of emotion or thought such as perplexity, disdain, skepticism, and mild amusement are observed, but fascinatingly, such nuance of expression is not seen when the same newborns are awake [12,13]. Smiles during REM sleep in newborns are highly frequent [9,57,58,59,60] and are considered to be endogenous and therefore not related to social experience [60]. They are therefore termed spontaneous smiles. Spontaneous smiles, as signs of immaturity, generally diminish and disappear at 2–3 months to be replaced by social smiles [61,62,63,64]. In some rare cases, however, spontaneous smiles can be observed even in 1-year-olds [65]. Nevertheless, smiling configuration during REM sleep often results in smiles with a closed mouth, whereas the smiles during wakefulness can be associated with an open mouth [52], indicating an inevitable social interactive impact.

Irregular breathing is always recorded in premature infants, while regular respiration, which characterizes NREM sleep in full-term newborns, is less recorded. Apnea is frequently observed during sleep in preterm and term infants [13,66]. Periodic breathing is abundant until 38 weeks of gestational age and then disappears in full-term infants. Contrary to the chest fluctuations during REM sleep, respiration during NREM sleep is regular. Respiratory patterns between REM and NREM sleep in full-term newborns are different. Remarkably, very irregular breathing constantly displays variation and is usually but not exclusively associated with rapid eye movements. During REM sleep, the respiratory rate is 18% greater than that during NREM sleep.

In full-term newborns, the mean frequency of the regular heart rate is 115–120 beats/min [67]. The heart rate is very rapid in prematures. The heart rate between REM and NREM sleep is as different as the disparity in respiratory rates; mean heart rate is 3.4% higher during REM sleep [12].

A cohort study of respiratory and heart rates during REM and NREM sleep across the first year of life shows that in infants from 1 month to >9 months of age, the mean respiratory rate during REM sleep decreases from 35.8 to 22.3 breath/min, whereas during NREM sleep, it reduces from 37.9 to 22.6 breath/min. The mean heart rate during REM sleep decreases from 134.7 to 110.8 beats/min, whereas during NREM sleep, it reduces from 132.1 to 107.8 beats/min [68].

### 2.3. REM Sleep Timing in Developmental Sleep-Wake Cycle

The emergent timing of sleep-wake cyclicity remains controversial until now. Different results are obtained according to different methodological approaches used [35,37]. By measuring rapid eye movements and EEG discontinuity, the sleep state cycle with a mean duration of 68 min is observed in a majority of neonates who are about 30 weeks of postconceptional age (PCA) [69]. The sleep cycle assessed by a motility monitoring system is found at 36 weeks of PCA, and the cycle length is approximately 60 min [34,70]. These data suggest that the ultradian biologic rhythm begins to be established in the early perinatal stage of brain development.

When the sleep–wake profile in full-term newborns is recorded using polysomnography for 4 h, Roffwarg et al. found that REM sleep appears soon after sleep begins, and the 1st sleep cycle has a shorter average duration than later cycles. The initial period of REM sleep is proportionately briefer than ensuing REM sleep periods, even though the lengths of sleep cycles are considered. The amount of REM sleep in the 1st cycle is approximately 1/2 of that in subsequent individual cycles. The mean duration of REM sleep prolongs almost threefold in the second cycle and tends to diminish slightly in the third cycle. Generally, the second and third cycles are split almost evenly between the REM and NREM sleep phases. The REM sleep percentage is fairly constant from the second cycle on. Thus, the mean duration of sleep cycles and mean length of REM sleep in newborns are respectively 52.9 and 25.4 min [12].

During the neonatal period, sleep onsets mostly begin with REM sleep, and the REM sleep episodes and those of NREM sleep alternate with a period of 50 to 60 min [12,71,72,73]. Within the first few weeks of life, though wakefulness involves a smaller proportion of time, and REM sleep involves a larger amount than in any other period of life, the total amount and percentage of REM sleep are diminished with increasing protracted intervals of wakefulness, particularly when locomotive capacity is attained [12,34].

The predictable appearance in the evening (around 20:00 h) of a long period of sleep, highly organized into REM-NREM sleep stages, occurs first in infants of 3 months of age [46]. Meanwhile, the appearance of NREM sleep at sleep onset and cyclic alternating patterns are sometimes observed [11,51]. Furthermore, NREM sleep is largely increased at night [50,73,74,75]. The circadian swing to the day–night cycle thus results from the consolidation of sleep-wake states and their finer coordination.

Between 4 1/2 and 6 months of age, REM sleep at sleep onset is brief and frequently interrupted by other stages or wakefulness. The amount of REM sleep decreases with age: as an infant matures, she shows less daytime REM sleep and sleep-onset REM sleep [46]. Additionally, the length of sleep cycles across the first year of age increases with age because of the proportional increase of NREM sleep [76]. Spindles and K complexes are fully formed by the ages of 3 and 6 months, respectively [39].

The initial REM period in children appears much later and is shorter than that in children who nap. Meanwhile, deep slow wave sleep extends and occupies the first hours. When children progressively approximate the diurnal pattern of uninterrupted daytime wakefulness, the 1st REM sleep period of the night usually appears 50 to 70 min after falling asleep, and REM sleep periods become longer towards the morning hours [12]. By 5 years old, daytime napping ceases and overnight sleep duration gradually declines throughout childhood, due to a shift to later bedtimes, with wake times remaining stable during the routine week [77]. After the age of 10, the sleep cycle lasts about 90–110 min as in an adult [39].

Collectively, the ontogenetic development of REM sleep in humans is summarized in Table 1, showing the developmental changes in amount, polysomnographic, behavioral, and physiological characteristics, and timing in the sleep-wake cycle of REM sleep from immature to mature.

## 3. Neurodevelopmental Disorders Associated with REM Sleep Disturbances during Early Development of Humans

### 3.1. Sudden Unexpected Infant Death (SUID), Sudden Infant Death Syndrome (SIDS)

Every year in the USA, approximately 3500 infants die suddenly and unexpectedly [78]. The sudden death of a baby less than 1 year old that is unexpected, unexplained, and with undetermined causes is labeled SUID [78]. The terminology reflecting the unexplained sudden death of an infant has been under discussion [78]. Namely, SIDS was first defined in 1969, and as a consequence, the majority of papers apply this terminology. SIDS victims are thought to succumb during sleep, commonly exhibit symptoms of asphyxia, and show signs of having been subjected to chronic hypoxia [79]. The majority of SUIDs, particularly in the SIDS literature, suggests that sudden death occurs mainly during a narrow developmental window of postnatal 1–6 months. This is a period when significant changes occur in sleep organization and in the maturation of the brainstem and cortical centers involved in cardiovascular, respiratory, and arousal state control [80,81,82]. In normal infants, irregular breathing and periods of apnea commonly appear during REM sleep [12]. Regular breathing occurs during periodic appearances of REM and NREM sleep episodes, intermixed with waking, following each other in succession throughout the sleep period.

Newborns at risk for SIDS have longer intervals between REM sleep epochs during the sleep cycle and a decreased tendency for short waking periods at 2 and 3 months of age [36]. It is known that the number of arousals during sleep in normal infants at 2–3 months old is greater than that in children at a mean age of 4.6 years. Spontaneous arousals occurred every 3–6 min in infants compared with 6–10 min in children [82,83]. These data indicate that the periodicity of sleep states in SIDS victims is disturbed and then results in a failure to arouse from sleep during a critical transient event, such as apnea, that might subsequently lead to death. Moreover, infants at risk for SIDS have an increased nighttime REM sleep that coincides with an early morning time period when most SIDS deaths occur, suggesting a link between disordered REM sleep and SIDS [84]. Therefore, the link between the peak occurrence of SIDS and the period of major sleep developmental changes suggests that SIDS might be state-related and could involve abnormal interactions between the state-modulated arousal threshold and central regulatory mechanisms of cardiovascular and respiratory control. Indeed, several hypothetical models exist, such as the triple risk [85], the quadruple risk, and the allostatic load model [86,87], with each highlighting a critical period. More recently, butyrylcholinesterase [88], an enzyme of the cholinergic system potentially providing a measure of autonomic (dys)function, has been suggested as a SIDS biomarker.

### 3.2. Narcolepsy

Narcolepsy is a neurological disorder characterized by excessive daytime sleepiness, cataplexy (sudden loss of muscle tone during waking), and loss of boundaries between sleep and wake, with frequent state transitions and intrusions of REM sleep into the other ongoing states [47,48,89,90]. It is estimated that the prevalence of narcolepsy ranges from 0.2 to 600 per 100,000 people in various countries [91]. Narcolepsy is due to a deficiency of hypothalamic hypocretin/orexin [92,93] likely following an autoimmune etiology [94]. More than 50% of the disease onsets occur in childhood before puberty [95], and the disease is often misdiagnosed with other neurological or psychiatric disorders such as epilepsy and attention deficit hyperactivity disorder (ADHD) [39]. Thus, the diagnosis is often delayed to a few years after the symptom onset [95]. In the International Classification of Sleep Disorders (ICSD) 3rd Edition, narcolepsy with and without cataplexy are divided into narcolepsy type 1 and narcolepsy type 2, respectively.

Cataplexy is often triggered by positive emotions, especially laughing. In addition, a recent cohort study found that during REM sleep, children with narcolepsy type 1 have more severe motor instability that emerges also from wakefulness (status cataplecticus). This motor instability occurring during REM sleep significantly affects subjective complaints of impaired nocturnal sleep and excessive daytime sleepiness [47]. Fortunately, although impaired by their sleep disorder, most children with narcolepsy develop quite normally, most likely by escaping the most critical period of brain development, i.e., before 2 years old. Indeed, narcolepsy occurs extremely rarely before age of 5 years.

### 3.3. Developmental Disabilities

The evolution of sleep architecture and sleep-wake organization in infants coupled with the development of systems critical to language, attention, and executive functions suggest that deficient REM sleep and disorders of sleep continuity could have a significant impact on infants and children by potentially altering the developmental trajectory of the brain [96]. Infants who suffer sleep fragmentation during the first year of life perform worse on executive functional tasks, have much more risk of poor language learning, and display less efficient attention processing in their later lives [97,98,99]. This suggests that early sleep fragmentation results in long-term consequences and that sleep potentially serves as a critical window for developmental disabilities.

A paucity of studies correlates REM sleep in developmental disabilities with degree of mental retardation. For example, the presence of autistic spectrum disorders (ASD) or Down syndrome is associated with fewer and briefer episodes of REM sleep [100]. Children with ASD show lower EEG beta activity during REM sleep over cortical visual areas compared with healthy controls [101]. This suggests that many of the cognitive profiles encountered in developmental disabilities could be a function of REM sleep deficits related to genetic anomalies or involvement of the ontogenetical brain regions susceptible to pathophysiological processes alike ASD.

Premature infants are at a higher risk for the development of cognitive delays and disabilities [102]. REM sleep with rapid eye movements is considered to reflect a more organized and mature CNS functioning as compared to REM sleep without them [103,104]. Premature neonates with more rapid eye movements during REM sleep have a better cognitive outcome at 6 months than those with less rapid eye movements [54].

Among infants with developmental disabilities of unknown etiology, higher REM sleep proportions of the TST are related to better motor, exploratory, social, eating, and intellectual outcomes [105], whereas less REM sleep has been found in mentally retarded subjects compared with typically developing controls [106]. More REM sleep without rapid eye movements characterizes infants with developmental delays and is found, for instance, among infants with intrauterine growth retardation [107]. Thus, REM sleep amount and the number of rapid eye movements during this sleep state might serve as a predictor of cognitive development above and beyond birth status and medical risk.

ADHD is a common neurodevelopmental disorder, affecting around 63 million children worldwide [108]. Sleep problems have been found in around 55% of children diagnosed with ADHD [109] and are associated with poorer cognitive and behavioral outcomes [110]. The core symptoms of ADHD are inattention, impulsivity, and general hyperactivity [111], which is associated with neurocognitive deficits [112]. Although numerous hypothetical models of ADHD exist, initially and based on behavioral observations, it is considered a disorder due to delayed structural brain maturation [113]. Compared with controls, one out of seven studies demonstrated lower REM sleep duration [114], whereas six observed a higher proportion [115,116,117,118]. Studies on sleep architecture and efficiency found that children with ADHD display shorter REM sleep latency in polysomnographic recording than typically developing children and have more subjectively reported sleep problems and daytime sleepiness levels [119]. When ADHD coexists with a tic disorder, the children with this comorbidity show not only shorter REM sleep latency but also an increased duration of REM sleep compared with healthy controls. Moreover, microarousals in light and REM sleep and short motor-related arousal occur during the sleep period in children with ADHD and comorbid children with ADHD and tic disorder [120]. Collectively, this may suggest that the pathophysiological mechanisms of ADHD could be closely related to REM sleep.

### 3.4. REM Sleep Behavior Disorder (RBD)

RBD is characterized by the resurgence of fetal and early postnatal motor activity patterns that closely resemble the early REM sleep of life. Several lines of evidence from basic research and documented clinical and video-polysomnographic findings in humans indicate that dysregulation of the developing sleep neuromotor system can be expected to have adverse long-term effects. For instance, behavioral and dream disturbances emerge during sleep as RBD later in the life of humans, dogs, and cats [121,122,123]. Neuromotor system dysfunction during REM sleep in the early development of life may thus have played an instrumental role in generating the long RBD prodrome leading up to the eventual emergence of clinical RBD [124].

Clinical characteristics of RBD are abnormal behaviors (i.e., sleep-related vocalizations or complex motor behaviors such as dream enactment, without the typical REM atonia) and EMG abnormalities during REM sleep are noted yet the literature on childhood is scant [125]. This parasomnia warrants in childhood a differential diagnosis with narcolepsy type 1 [126] and amongst other brainstem tumors. Approximately 0.5 to 1.25 percent in the general population may suffer from RBD [127].

Taken together, the key features of disordered REM sleep during early development presented in the neurodevelopmental disorders mentioned above are summarized and emphasized in Table 2.

## 4. REM Sleep in Early Development of Mammalia

The early sleep pattern of most animal models follows a similar evolution to that of humans in that they also have abundant REM sleep in early life, although postnatal differences can be noticed. The ontogenetic development of polysomnographic, behavioral, and physiological characteristics of REM sleep in many mammals displays similar features to those in humans [128,129]. Furthermore, the maturational stage of sleep patterns in neonatal animals appears to be well correlated with their central nervous system (CNS) maturity [28,130]. On the one hand, animals born with an immature CNS, such as the cat, rat, mouse, and rabbit, undergo considerable postnatal development of their sleep–wake patterns before an adulthood pattern is established. On the other hand, animals born with more advanced CNS maturation, such as the chimpanzee, monkey, and sheep, show sleep–wake patterns qualitatively and quantitatively similar to those of their adulthood. In animals born immature and during their early developmental phase, some authors suggested that sleep starts actually in a disorganized manner because of the indistinct cortical EEG activities [131,132]. Based on the muscle activity and behavioral criteria, AS and QS, i.e., early forms of REM and NREM sleep are also documented in altricial animals [80,130,133,134].

Table 3 summarizes the changes in the amount of REM sleep during early development in animal models. The data in Mammalia born immature clearly show a negative correlation between REM sleep amount along with the levels of postnatal development. Compared with mammals born immature, mammals born mature clearly show a lesser amount of REM sleep indicating that early brain development requires a greater amount of REM sleep. The function of REM sleep during early life would be to promote brain development which is in consistence with the ontogenesis hypothesis of REM sleep.

### Regulatory Mechanisms of Early REM Sleep Development

In humans, mice, rats and cats, REM sleep has generally been regarded as the most archaic state and is mediated by neural networks mainly located in the brainstem [73,80,150,151,152]. That is, the first cerebral structure to mature and the primary one to ensure basic cerebral and behavioral function from early life [153]. More recently, the REM sleep regulatory concept has been evaluated and several hypothalamic and forebrain networks including newly identified neuropeptides such as orexin and melanin-concentrating hormone (MCH) have been involved, both in the control and final expression of this behavioral state [6,27,154,155,156,157,158,159,160]. One of the most obvious correlations has been that when the human or animal brain at birth is less mature, the greater time spent in REM sleep in early postnatal life, such as REM sleep directly succeeding waking at sleep onset, is often observed in newborn humans [12,46,73] and mammals [16,18,20,22,23,138,152,161,162,163].

Concerning the brainstem mechanisms, several studies on rats suggest that cholinergic neurons of the laterodorsal (LDT) and pedunculopontine tegmentum (PPT) send projections to and activate glutamatergic neurons, with the pontine reticular formation to initiate and maintain REM sleep. However, serotonergic (5-HT) neurons within the dorsal raphe nuclei and noradrenergic neurons within the locus coeruleus project to the LDT and PPT to inhibit REM sleep [157]. Furthermore, several pontine and medullary areas that mediate muscle atonia and twitches during REM sleep in adults are also involved in the generation of these REM sleep components in the early development of life [30,164,165,166,167]. Brain neural structures responsible for REM sleep are therefore functional as early as pre- and postnatal stages and the appearance of adult-like NREM sleep requires cortical maturation.

The cerebral cortex of a newborn cat has a higher level of maturity than the newborn rat’s [168], which may explain why polysomnographic recording in the kitten immediately after birth shows an EEG activity that is not present in the newborn rat such as NREM sleep signs [18,169]. Rat cortical neurons between the birthday (P0) and P10 show explosive growth, cortical oscillations are only weakly modulated by behavioral states, and EEG activity is discontinuous. At around P11-P12, a pivotal cortical maturity transition occurs [130,170,171], coinciding at which NREM sleep appears [130].

Moreover, the effects of the numerous transmitter systems on the membrane potential of the neurons in the pedunculopontine nucleus in rats during the developmental decrease in REM sleep change, including increased 5-HT1 inhibition [172], decreased NMDA excitation [173], increased kainic acid activation [173], decreased noradrenergic inhibition [174], and increased cholinergic [175] and GABAergic inhibition [176]. These data suggest a reorganization of REM sleep-controlling neurons within the mesopontine tegmentum, such that the neuromodulation of REM sleep undergoes drastic changes from birth to the end of human puberty [177], i.e., from an early to mature modality.

During REM sleep, brainstem circuits actively suppress motor neurons in the spinal cord to keep skeletal muscle atonia [6,154,155,157,178]. Conversely, a marked amount of twitching and gross body movement is observed during REM sleep but not during NREM sleep during the early development of humans [12,13,34,35,38,52,53] and Mammalia [16,18,20,29,30,163,179,180]. This distinguished pattern of REM sleep in early life suggests that the inhibiting mechanisms of spinal cord motor neurons wired with the brainstem circuits are immature. On the other hand, strong motor excitation has to conquer tonic and phasic motor activity suppression, leading to muscle twitches during REM sleep in early life [11,181]. The origin of the excitatory drives to generate twitches during REM sleep is located in the brainstem of mice, rats, and cats [182,183,184], whereas the supraspinal drives that mediate motor suppression during REM sleep have their origin in the brainstem inhibitory centers of mice and rats [6,154,155,157]. Thus, excitatory and inhibitory brainstem outputs determine the occurrence of muscle twitches and gross body movements during REM sleep.

Collectively, the brainstem REM sleep circuits mature while increasingly interacting mutually/bidirectionally with those in the hypothalamic and forebrain across early development to coalesce REM sleep components, and consolidate REM sleep episodes, express sleep-wake ultradian and circadian rhythmicity.

## 5. Underlying Mechanisms Involved in Neurodevelopmental Disorders Associated with Early REM Sleep Disturbances in Mammalian Models

### 5.1. CNS Development

The primary function of REM sleep is proposed to be inducing the CNS development in the fetus as well as the neonates of humans, rats, cats, and guinea pigs [12,18,152] and constituting the major CNS stimulator in a period when waking life is limited in time and scope with the little occasion for stimulation in cats [142,185,186]. The functional stimulation commences in fetal life and may result not only from actual sensory stimulation but perhaps also from the REM sleep process, which starts to operate at some points in fetal development. The ascending impulses originating in the brainstem during REM sleep may be required in promoting neuronal differentiation, maturation, and myelination in higher brain centers as well as the maturation of the cardiorespiratory regulating center within the brainstem. Thus, the abundance of REM sleep in early life and its ensuing decline to lower levels in adulthood strongly suggest that REM sleep is an integral part of the activity-dependent processes that enable normal physiological and structural brain development in humans, rats, and cats [28,30,130,177,187]. Conversely, the early developmental deficiency of REM sleep and the neurons or neurotransmitters involved in brainstem circuits will cause neurodevelopmental disorders.

Increasing numbers of studies in animal models have provided the underlying mechanisms involved in some REM sleep-related disorders (Table 4). For example, REM sleep in postnatal rats is dramatically reduced throughout 2 weeks, and REM sleep-deprived rats in adulthood have reduced brain size, hyperactivity, anxiety, attention, and learning difficulties [28,82]. The ADHD-like behaviors and symptoms induced by REM sleep deprivation may be linked to decreased alpha2A-adrenoceptor signaling, particularly in the hippocampus [188]. When REM sleep deprivation in infant rats is carried out from P16 to P19 for 4 h per day, it reduces the stability of hippocampal neuronal circuits, possibly by hindering the expression of mature glutamatergic synaptic components that are involved in several neural processes such as brain maturation and memory consolidation [189], whereas an increase in REM sleep amounts induced by exposure to an enriched environment in the juvenile rat results in a significant increase in the adult brain weight, particularly the cerebral cortex and hypothalamus [190]. Similarly, REM sleep enhancement has also been reported in infant animals following learning tasks [191], suggesting that during the developmental period, the increased amount of REM sleep after a learning experience promotes brain growth.

### 5.2. Social Behaviors

REM sleep abnormalities in early life are prevalent in ASD. REM sleep reduction in early life causes long-lasting compensatory changes in GABAergic parvalbumin in the primary somatosensory cortex, impairments of pair bond formation, and alteration in object preference in adult prairie voles [192], suggesting that early life REM sleep is crucial for tuning inhibitory neural circuits and development of species-typical affiliative social behaviors. While sleep in male mice was deprived for 3 h per day from P5 to P52, these sleep-deprived mice displayed autistic-like behaviors including long-lasting hypoactivity and impaired social behavior in adolescence. These behavior changes were accompanied by an increase in the downstream signaling products of the mammalian target of the rapamycin pathway [193]. These shreds of evidence indicate that sleep deprivation can play a causative role in the development of behavioral abnormalities.

In addition, REM sleep deprivation in neonatal rats also induces depression-like behaviors in their adulthood, such as reduction of male sexual behaviors, pleasure-seeking, shock-induced aggression, REM sleep latency, and the enhancement of defensive responses, motor restlessness associated with the fear or stress, amount of REM sleep, voluntary alcohol consumption and despair behavior [194,195,196,197,198,199]. Thus, REM sleep appears to be closely related to emotional and mental development in early life.

### 5.3. SUID/SIDS

Several recent studies of animal models for underlying mechanisms involved in SUID/SIDS have found that neonatal Lmx1bf/f/p mice selectively lacked 5-HT neurons, displayed frequent and severe apnea, and had high mortality during early development. Excess mortality at the time of breathing abnormalities was the most severe [200]. While rat pups were deficient in central 5-HT, they were profoundly more apneic in REM sleep but not NREM sleep, and their arousal in hypoxia was delayed in REM sleep compared with NREM sleep [201]. Furthermore, in perinatal nicotine-exposed 5-HT-deficient rat pups, impaired autoresuscitation along with significantly delayed post-anoxic recovery of normal breathing and heart rate was observed at P10 [202]. These shreds of evidence indicate that the CNS 5-HT plays an important role in REM sleep and cardiorespiratory control, that infants who are deficient in central 5-HT may be at increased risk for SIDS in REM sleep because of increased apnea and delayed arousal, and that cigarette smoking during pregnancy increases the risk of SIDS also.

### 5.4. Narcolepsy

Narcolepsy, a neurodevelopmental disorder characterized primarily by REM sleep dysregulation, the animal model of prepro-orexin gene knockout mice exhibited a phenotype strikingly similar to human narcolepsy patients including hypersomnolence during their active dark phase, fragmented waking periods, SOREMS, [203] and cataplexy episodes [204]. In addition, orexin/ataxin-3 mice [205] and rats [206] were born with orexins but loose orexin-containing neurons later in life. Accordingly, the onset of narcoleptic attacks in orexin/ataxin-3 mice was later than in prepro-orexin knockout mice, which showed behavioral arrests, premature entry into REM sleep, poorly consolidated sleep patterns, and late-onset obesity [205]. The orexin/ataxin-3 rats exhibited postnatal loss of orexinergic neurons that resulted in the expression of a phenotype with fragmented vigilance states, a decreased latency to REM sleep, and increased REM sleep time during the dark active phase. SOREMS, a defining characteristic of narcolepsy, and cataplexy occurred frequently in these orexin/ataxin-3 transgenic rats [206]. These results provide evidence that orexin-containing neurons play important roles in regulating vigilance states and energy homeostasis.

**Table 4 ijerph-19-13101-t004:** Animal models for underlying mechanisms involved in the neurodevelopmental disorders associated with REM sleep disturbances.

Animal Models		Phenotypes	Underlying Mechanisms	Ref.
SIDS	c	Frequent and severe apnea, high mortality during development.	Selectively lack of 5-HT neurons induces abnormality of cardiorespiratory control.	[200]
	TPH2-/- rat pups	Increased apnea only in REM sleep. Arousal responses in hypoxia condition were selectively delayed in REM sleep.	Deficient in central 5-HT leads to a loss of inhibitory effect on LDT/PPT activation, and a failure in breathing.	[201]
	Perinatal nicotine-exposed 5-HT deficient rat pups	Autoresuscitation failure in response to hypoxia.	5-HT deficiency and perinatal nicotine exposure increase the vulnerability to environmental stressors and exacerbate defects in cardiorespiratory protective reflexes to repetitive anoxia during the development period.	[202]
Narcolepsy	Prepro-orexin gene KO mice	Hypersomnolence during the active phase, fragmented wakefulness, SOREMS, cataplexy.	Orexin deficiency fails to regulate the physiologic sleep-wake cycle.	[203,204]
	Orexin/ataxin-3 mice	Behavioral arrests, premature entry into REM sleep, poorly consolidated sleep patterns and obesity.	Postnatal loss of orexin fails to regulate vigilance states and energy homeostasis.	[205]
	Orexin/ataxin-3 rats	Fragmented vigilance states, decreased latency to REM sleep, and increased REM sleep time during the active phase, SOREMS and cataplexy.	The presence of orexin impacts vigilance state control through acting as a circadian arousal signal and inhibiting the SOREMS.	[206]
ASD	RSD in infant prairie voles	Impair pair bond formation and alter object preference in adulthood.	Early REM sleep is crucial for tuning inhibitory neural circuits and developing species-typical affiliative social behaviors.	[192]
	SD in infant mice from P5-P52	Long-lasting hypoactivity and impaired social behavior in adolescent.	Early sleep deprivation increases downstream signaling products of the mammalian target of rapamycin pathway.	[193]
ADHD	RSD in infant rats for 2 weeks	Reduced brain size, hyperactivity, anxiety, attention and learning difficulties.	Early REM deprivation damages brain maturation and cause ADHD-like behaviors.	[28,82]
	RED in infant rats	Memory deficit.	Reduction of stability of hippocampal neuronal circuits.	[189]
Depression	RSD in neonatal rats	Reduction of male sexual behaviors, pleasure-seeking, shock-induced aggression, REM sleep latency.	REM sleep promotes early emotional and mental development.	[197,198,199]

Abbreviation: Ref., references; RSD, REM sleep deprivation; SD, sleep deprivation; SOREMS, sleep-onset REM sleep.

## 6. Conclusions

The investigation of the ontogenetic development of REM sleep from humans to animals demonstrates that the appearance of recognizable REM sleep by EEG and its subsequently mechanistic maturation appears to follow a similar developmental program: REM sleep is remarkably abundant during the early period and declines progressively across development, and REM sleep ontogenesis presents a remarkably conserved feature of mammalian sleep. A core set of findings after multiple studies across species demonstrates that REM sleep in early life plays a critical role in the maturation and plasticity of the developing brain, physiology, and behaviors. Conversely, if REM sleep is deficient, significant changes occur in sleep organization and the maturation of the brainstem and cortical centers; cardiovascular and respiratory control may be jeopardized and neurodevelopment disorders may occur such as SUID/SIDS, narcolepsy, developmental disabilities, and various forms of mental retardation are increased. Based on these findings, the neurological mechanisms and functions of REM sleep involved in the drastic change from immature to mature modality and neurodevelopmental disorders require future in-depth studies. Further assessment of the relationship between early life REM sleep and the developing brain is necessary for preventing and treating these disorders. Similarly, the progress of research technology and the continuous improvement of EEG data collection during early human life are also important directions for improved understanding of REM sleep and its role in mammalian development.

## Figures and Tables

**Table 1 ijerph-19-13101-t001:** Developmental changes of amount, polysomnographic, behavioral, and physiological characteristics, and timing in the sleep-wake cycle of REM sleep across early development of humans.

Parameters		REM Sleep	References
Amount	Preterm	80% of TST at 30 GW, 67% between 33 and 35 GW, 58% between 36 and 38 GW.	[13,31,32]
	Full-term	50% of TST in full-term newborns.	[9,10,11,12,38]
	Postnatal	Progressive reduction with age, reaching 20% of TST at about three years of age and remaining constant throughout childhood, adolescence, and adulthood.	[9,11,12,39,40]
EEG	Preterm	Inconstant EEG of REM sleep is surveyed <30 GW, a constant pattern is observed during 36–38 GW. EEG remains atypical.	[13,31,34,37,42]
	Full-term	Easy identification with low-voltage, relatively fast activities, frequent occurrence of REM sleep directly succeeds waking episodes at sleep onset.	[12,35,43,44]
	Postnatal	EEG patterns progressively increase in frequency and amplitude. The occurrence of sleep beginning with REM sleep declines with age, from 60% at 3 weeks to 20% at 6 months.	[49]
Rapid eye movements (REMS)	Preterm	Eye movements are very rare <28 GW. number of REMS remains lower.	[35,54]
	Full-term	More REMS, EOG invariably appears as single or clustered high-amplitude bursting waves.	[43,44]
	Postnatal	REMS starts to increase after birth, reaching a plateau at about 4 months.	[55]
Spontaneous body movements	Preterm	A large number between 38 and 40 GW, but the amount is lesser than that in full-term newborns.	[53]
	Full-term	Atonia becomes obvious. Grimaces, small weak cries, smiles, and twitches of the face and extremities are frequently observed. EMG shows phasic muscular contractions in the background of the absence of resting muscle activity.	[12,13]
	Postnatal	Spontaneous smiles generally diminish and disappear at 2–3 months and are replaced by social smiles. Spontaneous body movements decline with age.	[62,64]
Breathing	Preterm	Irregular, frequent apnea, periodic breathing <38 GW.	[13,66]
	Full-term	The respiratory rate during REM sleep is 18% greater than that during NREM sleep. Frequent apnea.	[12]
	Postnatal	For infants from 1 month to >9 months of age, the mean respiratory rate during REM sleep decreases from 35.8 to 22.3 breath/min	[68]
Heart rate	Preterm	Irregular, 130 beats/min at 37 GW.	[13,35]
	Full-term	115–120 beats/min, the mean heart rate is 3.4% higher during REM sleep than during NREM sleep.	[12,67]
	Postnatal	For infants from 1 month to >9 months of age, REM sleep decreases from 134.7 to 110.8 beats/min	[68]
Sleep-wake cycle	Preterm	Approximately 60 min.	[34,69]
	Full-term	The mean duration of sleep cycles and mean length of REM sleep in newborns are respectively 52.9 and 25.4 min. The amount of REM sleep in the 1st cycle is approximately 1/2 of that in subsequent individual cycles. The mean duration of REM sleep prolongs almost threefold in the 2nd cycle and tends to diminish slightly in the 3rd cycle.	[12]
	Postnatal	Length of sleep cycles across the first year of age is progressively increasing with age. REM sleep periods become longer towards the morning hours. After the age of 10, the sleep cycle lasts about 90–110 min as in an adult.	[12,39,76]

**Table 2 ijerph-19-13101-t002:** Disordered REM sleep features during early development related to neurodevelopmental disorders in humans.

Diseases	Onset Period	Disordered REM Sleep	References
SUID/SIDS	Infant (1–6 months)	Longer intervals between REM sleep epochs during the sleep cycle and a decreased tendency for short waking periods. Failure to arouse from sleep during a critical transient event, such as apnea. An increased nighttime REM sleep coincides with an early morning time period.	[36,84]
Narcolepsy	Childhood	Intrusions of REM sleep into the other ongoing states. Narcolepsy Type 1 has more severe motor instability during REM sleep.	[47,48,90]
ASD	Childhood	Fewer and briefer episodes of REM sleep. Lower EEG beta activity during REM sleep over cortical visual areas.	[100,101]
Prematurity	Infant	REM sleep with less or without REMS. Less REM sleep.	[54,106]
ADHD	Childhood	Shorter REM sleep latency and more daytime sleepiness. ADHD coexists with tic disorder showing not only shorter REM sleep latency but also an increased duration of REM sleep. Microarousals and short motor-related arousal during REM sleep.	[119,120]
RBD	Adulthood	Neuromotor system dysfunction during REM sleep in early development.	[124]

**Table 3 ijerph-19-13101-t003:** Changes in amount of REM sleep during early development in animals.

Animals		REM Sleep	References
Animals born with advanced maturation	Chimpanzee	22.4% of TST < 1 year, 16.0% between 1 and 2 years, and 13.1% above 2 years old.	[135]
	Rhesus monkey	31% of TST at birth, a brief increase to 43% at day 7, then, gradually decreases to 35% at day 30, to 27% between 9 and 13 months, and 19% (15% to 23%) at 2 years old.	[136,137,138,139,140]
	Sheep	60% of TST at 120 days of gestation, 45% at birth, 18% at day 7, and 14.71% at day 15.	[23,141]
Animals born with immaturity	Kitten	In its first days, 50% in REM sleep (100% of TST) and 50% in wakefulness. 50% of total recording time (TRT) on day 7, and 20% on day 35.	[18,142,143,144,145,146,147]
	Rat	72% of TRT in the first week, 58% at day 11, 8% at day 30.	[16,18]
	Mouse	40% of TRT in the first week, 6% at day 19.	[148]
	Rabbit	75% of TST at birth, 33% on day 14, and 10% on day 23.	[149]

## Data Availability

Not applicable.

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
