# Peer review of "Rapid Eye Movement Sleep during Early Life: A Comprehensive Narrative Review"

_ijerph, 2022, doi:10.3390/ijerph192013101_

Round 1

Reviewer 1 Report

Dear authors,

thank you very much for your manuscript. I have detected some spelling mistakes and highlighted them in the attached manuscript.

Best wishes

Author Response

Point 1: Dear authors, thank you very much for your manuscript. I have detected some spelling mistakes and highlighted them in the attached manuscript.

Response 1: We thank the reviewer for the encouraging comment. The spelling mistakes that the reviewer detected were revised as follows:

-Line 324, “qualitatively and qualitatively” replaced by “qualitatively and quantitatively”.

-Line 335, “be then” replaced by “be”.

-Line 366, “right moment” replaced by “right period”.

-Line 485, “Conversely, REM sleep is deficient while the significant changes occur in sleep organization…” is changed to “Conversely, if REM sleep is deficient, the significant changes occur in sleep organization…”

Reviewer 2 Report

This article is a narrative review of REM sleep during early life. The authors have integrated findings from both humans and animals to illustrate the importance of REM sleep for brain functions and development during the early life. The sections on human REM sleep were well-written, but could benefit from including more discussions on the biological/neuronal mechanisms. For the sections on animal REM sleep, the authors should always specify which animal models were described/discussed. My other comments are below:

1. Abstract: The first sentence of abstract discussed the ontogenetic sleep hypothesis, which was never mentioned in the main text of the manuscript. The authors should either remove this description in the abstract or discuss it in the main text.

2. The definition of "early life" can be quite broad. I suggest the authors define "early life" at the beginning of the article to provide readers with a context of what development stages are being discussed.

3. The last sentence of section 1 does not include what was discussed in section 2 (characteristics of REM sleep in early life). "This review focuses on some REM sleep-related disorders in humans and their underlying mechanisms as examined in animals." 

I suggest the authors revise this sentence.

4. Line 68, should "one-half of their sleep" be "one-half of their night"?

5. Line 70-73. The proportion of REM sleep does not stay constant. It does decline slightly from early througout late adulthood (see Ohayon et al., 2004 meta analysis).

6. Line 78. I'm not sure what "inconstantly notified" means. Please clarify.

7. Section 2.2.2 discussed behavioral features (eye movements, muscle contractions, etc.) and Section 2.2.3 discussed physiological features (respiration, heart rate). I'm not sure what criteria were used to categorize behavioral and physiological features (e.g., one could argue that muscle contraction is a physiological feature or breathing rate is a behavioral feature). 

8. Line 142 "The irregular heart rate is very rapid in prematures." This sentence is confusing, please clarify.

9. Line 150-151. The previous sentences do not lead to the conclusion that "the respiratory patterns and heart rate in early life are similar to that in adults." The reported breathing and heart rates in infants are much faster than those in adults.

10. Line 185. I'm not sure what "ordered REM sleep" means. Please clarify.

11. Line 200 and line 203. The term "evolutional changes" is not appropriate here. "Developmental changes" would be a more accurate description.

12. Section 3.1. The authors used two terms (i.e., SUIDS and SIDS). What are the differences between these two diseases? Please clarify. If they are referring to the same thing, please use one of the terms throughout the manuscript for consistency.

13. Section 3.1. I suggest the authors define SUIDS/SIDS and report the prevalence of this disease. It would be helpful for readers who are not familiar with this disease.

14. Line 221 "These data indicate that the periodicity of sleep states in SIDS victims is disturbed and then results in a failure to arouse from sleep during a critical transient event, such as apnea, that might subsequently lead to death." The previous sentences described REM sleep characteristics of the victims of SIDS, which indicate an association, but not causal relationship. Therefore, the authors' conclusion is not accurate. The cause of SIDS should be the compromised autonomic regulation for cardiovasuclar and/or respiratory mechanisms (which were only briefly mentioned in the last sentence of this section). 

15. Section 3.2. The authors did a great job defining narcolepsy. It would be helpful if the authors could report the prevalence of narcolepsy for readers who are not familiar with it.

16. Line 295, "This suggests that the ADHD pathophysiological mechanisms underpinning hypermotor symptoms..." The authors have not discussed the biological/neuronal mechanisms in this section, so I'm not sure how they made this conclusion.

17. In general, I think sections 2 and 3 will be substantially improved if the authors also discuss the underlying biological/ neuronal/ pathological mechanisms.

18. In sections 4 and 5, it is unclear whether some sentences are describing humans or animals. If the latter, the authors should specify which animal is discussed (e.g., rat, cat, dog?). E.g.,

- line 339-345

- line 350-359

- line 368-375

- line 380-392

- line 396-409

It also seems that only mammals were discussed in sections 4 and 5. If so, this should be made clear early in the manuscript.

19. Line 324: "...show sleep-wake patterns qualitatively and qualitatively similar.." Should one of them be quantitatively?

20. Line 360-361: The authors compared newborn cat with newborn rat. This comparison is not appropriate. And because this is a cross-species comparison, many factors other than brain maturation level could explain "why polysomnographic recording in the kitten immediately after birth shows an EEG activity that is not present in the newborn rat". Also, the authors described that newborn cats had a highly development maturation of cerebral cortex, this is contrary to the description on line 320 that cats are born with immature CNS. 

21. Line 479. I'm not sure what "experimentally identified REM sleep" means. Please clarify.

22. The sentence on line 485-489 seems to suggest that REM sleep disturbance caused disturbances in cardiovascular and respiratory control. This causal relationship is not supported by any evidence presented in this manuscript.

23. After a thorough review of the literature, it would be extremely beneficial if the authors could discuss the specific gaps in the literature and discuss specific future directions for researchers in this field. Therefore, I suggest the authors elaborate on future directions. 

Author Response

Point 1: This article is a narrative review of REM sleep during early life. The authors have integrated findings from both humans and animals to illustrate the importance of REM sleep for brain functions and development during the early life. The sections on human REM sleep were well-written, but could benefit from including more discussions on the biological/neuronal mechanisms. For the sections on animal REM sleep, the authors should always specify which animal models were described/discussed. My other comments are below:

We are very appreciative of your comments and we hope to have satisfactorily addressed them. Thank you for your time reading our work.

Point 1. Abstract: The first sentence of abstract discussed the ontogenetic sleep hypothesis, which was never mentioned in the main text of the manuscript. The authors should either remove this description in the abstract or discuss it in the main text.

Response 1: We thank the reviewer for the encouraging comment. A new sentence, “The ontogenetic sleep hypothesis pointed out that the dramatic decline in REM sleep amounts across development manifests that REM sleep ontogenesis is a remarkably conserved feature of mammalian sleep [28], which suggested that the REM sleep is ontogenetically primitive.”, is inserted into the Line 53-57. In addition, animal models are also explained in the updated manuscript.

Point 2: The definition of "early life" can be quite broad. I suggest the authors define "early life" at the beginning of the article to provide readers with a context of what development stages are being discussed.

Response 2: Thank you for capturing this oversight. We defined it at line 46-48 in the revised manuscript.

Point 3: The last sentence of section 1 does not include what was discussed in section 2 (characteristics of REM sleep in early life). "This review focuses on some REM sleep-related disorders in humans and their underlying mechanisms as examined in animals." I suggest the authors revise this sentence.

Response 3: Thank you for this comment. This sentence is changed to “This review focuses on the characteristics of REM sleep in early life, some REM sleep-related disorders in humans and their underlying mechanisms as examined in animals.”

Point 4: Line 68, should "one-half of their sleep" be "one-half of their night"?

Response 4: Thank you for this remark. It means that the REM sleep accounts for half of the total sleep time. Roffwarg et al illustrated that the first 15 days of life the neonate spends proportionally an equal amount of their sleep time in REM and NREM sleep.

Point 5: Line 70-73. The proportion of REM sleep does not stay constant. It does decline slightly from early throughout late adulthood (see Ohayon et al., 2004 meta analysis).

Response 5: This sentence is changed to “Later on, the percentage in REM sleep over total sleep time (TST) progressively declines with age and reaches a roughly stable proportion of 20% in REM sleep and 80% in NREM sleep at about the age of three years to remain approximately constant throughout childhood, adolescence, and early adulthood [9,11,12,39,40], whereas in late adulthood, it does decline slightly [41].”, and the reference “Ohayon et al., 2004” is cited as reference [41] in the updated manuscript. We have placed more nuances because we agree that more objective data and/or reference values are needed.

Point 6: Line 78. I'm not sure what "inconstantly notified" means. Please clarify.

Response 6: Thank you for this comment. We intended to express that before 30 weeks of gestation, the characteristics of the EEG are not clear, that is, it is difficult to determine which state the fetus is in. This sentence is changed to “In humans, sleep and electroencephalographic (EEG) patterns of REM and NREM sleep can be difficult to decipher before 30 weeks of gestation.”

Point 7: Section 2.2.2 discussed behavioral features (eye movements, muscle contractions, etc.) and Section 2.2.3 discussed physiological features (respiration, heart rate). I'm not sure what criteria were used to categorize behavioral and physiological features (e.g., one could argue that muscle contraction is a physiological feature or breathing rate is a behavioral feature).

Response 7: Thank you for pointing this out, we agree and we reorganized these two sections into “2.2.2. Behavioral and physiological features”.

Point 8: Line 142 "The irregular heart rate is very rapid in prematures." This sentence is confusing, please clarify.

Response 8: This sentence is changed to “The heart rate is very rapid in prematures.”

Point 9: Line 150-151. The previous sentences do not lead to the conclusion that "the respiratory patterns and heart rate in early life are similar to that in adults." The reported breathing and heart rates in infants are much faster than those in adults.

Response 9: We agree and have deleted this sentence.

Point 10: Line 185. I'm not sure what "ordered REM sleep" means. Please clarify.

Response 10: Thank you for this remark, we agree. That is, the adjective “ordered” makes the sentence difficult to understand, therefore, this word has been deleted.

Point 11: Line 200 and line 203. The term "evolutional changes" is not appropriate here. "Developmental changes" would be a more accurate description.

Response 11: We thank for the reviewer’s suggestion, “evolutional” replaced by “developmental”.

Point 12: Section 3.1. The authors used two terms (i.e., SUIDS and SIDS). What are the differences between these two diseases? Please clarify. If they are referring to the same thing, please use one of the terms throughout the manuscript for consistency.

Response 12: Per the Centers for Disease Control and Prevention (CDC) SUID is the “Sudden Unexpected Infant Death describing the sudden and unexpected death of a baby less than 1 year old in which the cause was not obvious before investigation. These deaths often happen during sleep or in the baby’s sleep area’’. The term SIDS or Sudden Infant Death Syndrome was first defined by Dr. Bruce Beckwith in 1969, “the sudden death of any infant or young child, which is unexpected by history, and in which a thorough post-mortem examination fails to demonstrate an adequate cause for death”. Over the years there has been some debate about the terminology due to the term “syndrome” in SIDS, particularly by medical examiners and coroners but also lawyers because a syndrome implies a disease or condition with a common group of signs and symptoms. The umbrella term SUID where U can stand for unexplained, unexpected, and undetermined causes was later purposed. As a consequence, terms like SIDS, SUID, SUDI, sleep-related infant deaths, etc … are used interchangeably. For more on the discussion we respectfully refer you to Shapiro-Mendoza CK, Palusci VJ, Hoffman B, Batra E, Yester M, Corey TS, Sens MA; AAP TASK FORCE ON SUDDEN INFANT DEATH SYNDROME, COUNCIL ON CHILD ABUSE AND NEGLECT, COUNCIL ON INJURY, VIOLENCE, AND POISON PREVENTION, SECTION ON CHILD DEATH REVIEW AND PREVENTION, NATIONAL ASSOCIATION OF MEDICAL EXAMINERS. Half Century Since SIDS: A Reappraisal of Terminology. Pediatrics. 2021 Oct;148(4):e2021053746. doi: 10.1542/peds.2021-053746. Epub 2021 Sep 20. PMID: 34544849; PMCID: PMC8487943.

Because of the discussion on the SUID/SIDS terminology it is difficult to apply one term across the manuscript. We therefore chose to apply the terminology used by the cited authors.  

Point 13: Section 3.1. I suggest the authors define SUIDS/SIDS and report the prevalence of this disease. It would be helpful for readers who are not familiar with this disease.

Response 13: Thank you for your comment. We have added the information for the USA.

Point 14: Line 221 "These data indicate that the periodicity of sleep states in SIDS victims is disturbed and then results in a failure to arouse from sleep during a critical transient event, such as apnea, that might subsequently lead to death." The previous sentences described REM sleep characteristics of the victims of SIDS, which indicate an association, but not causal relationship. Therefore, the authors' conclusion is not accurate. The cause of SIDS should be the compromised autonomic regulation for cardiovasuclar and/or respiratory mechanisms (which were only briefly mentioned in the last sentence of this section).

Response 14: Thank you for your comment. Mostly SIDS papers reported on REM sleep. We however agree that several hypothetical models exist, in addition to the discussion regarding the terminology, more research is needed to clarify the exact susceptibility of perturbed sleep in SUID/SIDS. It has long been believed that the failure to arouse from sleep apnea was the cause of SIDS, hence, at risk infants were send home with an cardio-respiratory monitoring system which alerted the parents. Please note, similar to the other neurodevelopmental problems discussed, SUID/SIDS is used as an illustration with respect to REM sleep.

Point 15: Section 3.2. The authors did a great job defining narcolepsy. It would be helpful if the authors could report the prevalence of narcolepsy for readers who are not familiar with it.

Response 15: “It is estimated that the prevalence of narcolepsy ranges from 0.2 to 600 per 100,000 people in various countries [85].” has been inserted into the updated manuscript.

Point 16: Line 295, "This suggests that the ADHD pathophysiological mechanisms underpinning hypermotor symptoms..." The authors have not discussed the biological/neuronal mechanisms in this section, so I'm not sure how they made this conclusion.

Response 16: This sentence is replaced by “Collectively, this may suggest that the pathophysiological mechanisms of ADHD could be closely related to REM sleep.”

Point 17: In general, I think sections 2 and 3 will be substantially improved if the authors also discuss the underlying biological/ neuronal/ pathological mechanisms.

Response 17: Although we agree with the reviewer, unfortunately adding more would defer us from the main scope of reviewing REM sleep (across species). Each of the section 2 and 3 points could be subjected to a separate review paper, and we share with the reviewer the enthusiasm on the topic but for the current paper we limited it to an illustrative approach (current paper has xxxx words). That is, we hope that our narrative review will stimulate more research on the ontogenetic sleep hypothesis, which has been neglected in the last decades.

Point 18: In sections 4 and 5, it is unclear whether some sentences are describing humans or animals. If the latter, the authors should specify which animal is discussed (e.g., rat, cat, dog?). E.g.,

- line 339-345

- line 350-359

- line 368-375

- line 380-392

- line 396-409

It also seems that only mammals were discussed in sections 4 and 5. If so, this should be made clear early in the manuscript.

Response 18: Thank you, we have made modifications in the revised manuscript.

Point 19: Line 324: "...show sleep-wake patterns qualitatively and qualitatively similar.." Should one of them be quantitatively?

Response 19: Thank you for capturing this oversight. We have been changed it.

Point 20: Line 360-361: The authors compared newborn cat with newborn rat. This comparison is not appropriate. And because this is a cross-species comparison, many factors other than brain maturation level could explain "why polysomnographic recording in the kitten immediately after birth shows an EEG activity that is not present in the newborn rat". Also, the authors described that newborn cats had a highly development maturation of cerebral cortex, this is contrary to the description on line 320 that cats are born with immature CNS.

Response 20: Thank you for pointing out this inconsistency. This sentence is changed to “The cerebral cortex of a newborn cat has a higher level of maturity than the newborn rat’s…”.

Point 21: Line 479. I'm not sure what "experimentally identified REM sleep" means. Please clarify.

Response 21: We intended to express the REM sleep which can be distinguished by EEG. This sentence is changed to “…that the appearance of recognizable REM sleep by EEG and its subsequently mechanistic…”

Point 22: The sentence on line 485-489 seems to suggest that REM sleep disturbance caused disturbances in cardiovascular and respiratory control. This causal relationship is not supported by any evidence presented in this manuscript.

Response 22: We agree, and thank you for this comment. This sentence is replaced by “Conversely, if REM sleep is deficient, the significant changes occur in sleep organization and in the maturation of the brainstem and cortical centers, therefore cardiovascular and respiratory control may be jeopardized and neurodevelopment disorders may occur, such as SUID/SIDS, narcolepsy, developmental disabilities, and various forms of mental retardation are increased.”

Point 23: After a thorough review of the literature, it would be extremely beneficial if the authors could discuss the specific gaps in the literature and discuss specific future directions for researchers in this field. Therefore, I suggest the authors elaborate on future directions.

Response 23: In the revised manuscript, the last three sentences of the discussion elaborate on this.

Reviewer 3 Report

This review focuses on ontogenetic development of REM sleep in both humans and animals, and is a valuable works. It systematically summarizes the characteristics and associated diseases of REM sleep during early life, and provides a detailed account of the physiological and behavioral changes during REM sleep development and contributes to our understanding of basic research and clinical phenomena in REM sleep-related diseases. Overall, the article is well-written and citation is appropriate. There are some modifications need the authors to address in the revised manuscript. 

1.       Page 1 line 21-23, “In coincidence with a … across species.” The comma within the sentence should be omitted.

2.       Page 1 line 27, when “ADHD” first appears, it should be explained.

3.       Page 3 line 104-105, “those whose sleep at 6 months begins with REM sleep reduced to 20% ”.  Suggest “… are reduced to 20%”.

4.       Table 4. Narcolepsy, Orexin/ataxin-3 mice, a period “.” should be added to the end of the sentence “Behavioral arrests…”

5.       Table 4 should be more concise and avoiding large paragraphs.

6.      The paragraphs in all tables should be rearranged for easy reading, for example, to decrease the front size and justify left.

Author Response

Point 1: This review focuses on ontogenetic development of REM sleep in both humans and animals, and is a valuable works. It systematically summarizes the characteristics and associated diseases of REM sleep during early life, and provides a detailed account of the physiological and behavioral changes during REM sleep development and contributes to our understanding of basic research and clinical phenomena in REM sleep-related diseases. Overall, the article is well-written and citation is appropriate. There are some modifications need the authors to address in the revised manuscript.

Thank you for your time to read our work, and we greatly appreciated the constructive comments.

Point 1. Page 1 line 21-23, “In coincidence with a … across species.” The comma within the sentence should be omitted.

Response 1: We thank the reviewer for the encouraging comment. We have deleted the comma and this sentence is replaced by “In coincidence with a rapidly developing brain during the early period of life a remarkably large amount of REM sleep has been identified in numerous behavioral and polysomnographic studies across species.”

Point 2: Page 1 line 27, when “ADHD” first appears, it should be explained.

Response 2: Thank you for capturing this oversight. “ADHD” is changed to “attention deficit hyperactivity disorder (ADHD)”

Point 3: Page 3 line 104-105, “those whose sleep at 6 months begins with REM sleep reduced to 20% ”.  Suggest “… are reduced to 20%”.

Response 3: We thank the reviewer’s suggestion, "…REM sleep reduced to 20%” is changed to “…REM sleep are reduced to 20%”.

Point 4: Table 4. Narcolepsy, Orexin/ataxin-3 mice, a period “.” should be added to the end of the sentence “Behavioral arrests…”

Response 4: Thank you, we have modified Table 4 accordingly.

Point 5: Table 4 should be more concise and avoiding large paragraphs.

Response 5: Thank you for this comment, a new Table 4 has been uploaded in the manuscript.

Point 6: The paragraphs in all tables should be rearranged for easy reading, for example, to decrease the front size and justify left.

Response 6: Thank you for this comment we have modified to the best, and potentially the journal will further implement their housestyle.

Round 2

Reviewer 2 Report

The authors have addressed all my previous comments. I support the publication of this manuscript.